# Phenotypic and Genomic Comparison of *Staphylococcus aureus* Highlight Virulence and Host Adaptation Favoring the Success of Epidemic Clones

Fengning Chen,[a,b] Yuyao Yin,[b] Hongbin Chen,[b] Longyang Jin,[b] Shuguang Li,[b] Ruobing Wang,[b] Shuyi Wang,[a] Qi Wang,[b] Shijun Sun,[b] Hui Wang[a,b]

[a]Institute of Medical Technology, Peking University Health Science Center, Beijing, China
[b]Department of Clinical Laboratory, Peking University People's Hospital, Beijing, China

**ABSTRACT** Methicillin-resistant *Staphylococcus aureus* (MRSA) of the sequence type 59 (ST59) and ST398 lineages has emerged in hospitals and displayed a higher virulent potential than its counterparts ST5 and ST239. However, the mechanism of the host cell-pathogen interaction and specific determinates that contribute to the success of epidemic clones remain incompletely understood. In the present study, 142 *S. aureus* strains (ST59, ST398, ST239, and ST5) were selected from our 7-year national surveillance of *S. aureus* bloodstream infections ($n = 983$). We revealed that ST59 and ST398 had a higher prevalence of the protease-associated genes *hysA^{VSaβ}*, *paiB*, and *cfim* and enhanced proteolytic activity than the other lineages. ST59 and ST398 showed a higher expression of *RNAIII* and *psmα* and greater proficiency at causing cell lysis than other lineages. Furthermore, ST59 and ST398 were strongly recognized by human neutrophils and caused more cell apoptosis and neutrophil extracellular trap degradation than the other lineages. In addition, these strains differed substantially in their repertoire and composition of intact adhesion genes. Moreover, ST398 displayed higher adaptability to human epidermal keratinocytes and a unique genetic arrangement inside the oligopeptide ABC transport system, indicating functional variations. Overall, our study revealed some potential genomic traits associated with virulence and fitness that might account for the success of epidemic clones.

**IMPORTANCE** Considerable efforts have been exerted to identify factors contributing to the success of epidemic *Staphylococcus aureus* clones, however, comparative phenotypic studies lack representation owing to the small number of strains. Large-scale strain collections focused on the description of genomic characteristics. Moreover, methicillin-resistant *S. aureus* infections constitute 30% to 40% of *S. aureus* bloodstream infections, and recent research has elucidated highly virulent methicillin-susceptible *S. aureus* strains. However, comprehensive research on the factors contributing to the success of epidemic *S. aureus* clones is lacking. In this study, 142 *S. aureus* strains were selected from our 7-year national surveillance of *S. aureus* bloodstream infections ($n = 983$) accompanied by a rigorous strain selection process. A combination of host cell-pathogen interactions and genomic analyses was applied to the represented strains. We revealed some potential genomic traits associated with virulence and fitness that might account for the success of epidemic clones.

**KEYWORDS** *Staphylococcus aureus*, lineage replacement, neutrophil, adhesion and invasion, virulence determinants

Address correspondence to Hui Wang, whuibj@163.com.

The authors declare no conflict of interest.

*S*taphylococcus aureus is a human skin and mucosae commensal but also a notorious pathogen that causes infections ranging from mild skin infections to fatal diseases (1, 2). Although there is great population diversity in *S. aureus* epidemiology,

relatively few methicillin-resistant *Staphylococcus aureus* (MRSA) clones dominate in specific regions. However, the predominant status is not permanent because clonal replacement occurs dynamically during the evolutionary process (3–5). Furthermore, community-associated *S. aureus* has emerged as a significant cause of nosocomial infections worldwide (1). Considerable efforts have been made to reveal the clonal shift phenomenon in China, indicating a dramatic decrease in sequence type 239 (ST239) and the emergence of ST59 (previously recognized as community associated *S. aureus* [CA-*S. aureus*]) and ST398 in hospitals (6–9). Although the reason for this dynamic nature of the clonal *S. aureus* population remains uncertain, it is recognized that the success of epidemic MRSA clones is multifactorial, with phenotypic and genomic traits involved (10, 11).

A major contributor to the success of *S. aureus* as a pathogen is the abundance of cytotoxins and exoenzymes. Cytotoxins act on host cell membranes, resulting in the lysis of target cells and inflammation, whereas proteases play roles in nutrient acquisition, bacterial dissemination, and immune evasion (12). Previous studies have shown the significant contribution of the *agr* system to the pathogenesis of ST59 CA-MRSA compared with ST239-MRSA, especially in terms of cytolytic ability (13, 14). We previously found that the chemotaxis inhibitory protein (*chp*) was a strong candidate for the increased cytolytic potential for ST59 (13); the mechanism behind *chp* and cytolytic ability is also under investigation. However, the hemolysis and proteolysis abilities among *S. aureus* clones and the lineage-specific genes that might account for the difference have not been adequately investigated.

In addition to *in vitro* virulence, advantages in host-pathogen interactions also contribute to the success of epidemic clones (11). Baldan et al. (15) showed that ST22 had a significant higher capacity to invade A549 cells than ST228. Brazilian ST1-SCCmecIV strains had adapted to hospitals by gaining the ability to persist and survive inside host cells (16). Different effects on osteoblasts were observed for four Italian *S. aureus* clones, as follows: ST239, ST5, ST22, and ST228 (17). Interaction with keratinocytes is a critical step in the colonization of human skin, and the innate immune system represents the first line of defense against systemic *S. aureus* infections. Phagocytes, especially neutrophils, act like primary infection determinants (18). Neutrophil extracellular trap (NETs) formation is involved mainly in host defense during *S. aureus* infection. Hence, it is essential to determine whether different *S. aureus* clones display different behaviors while interacting with human epidermal keratinocytes and neutrophils, and the virulence determinates that are associated with these phenotypes.

Altogether, there is a lack of comprehensive research on the factors that contribute to the success of epidemic *S. aureus* clones. In this study, 142 *S. aureus* strains (ST59-MRSA/MSSA, ST398-MRSA/MSSA, ST239-MRSA, and ST5-MRSA/MSSA) were selected from our 7-year national survey of *S. aureus* bloodstream infections ($n = 983$; 2013 to 2020) accompanied by a rigorous strain selection process. A combination of phenotypic (proteolytic, cytolytic, and host cell-pathogen interactions) and genomic analyses (comparative genomic and mobile genetic element [MGE] analyses) were applied to the represented strains. As a result, we revealed some potential genomic traits associated with virulence and fitness phenotypes that might account for the success of epidemic clones.

## RESULTS

**Phylogeny constructions of the epidemical clones ST59 and ST398.** A total of 983 *S. aureus* isolates were collected from 13 cities in China. We found that ST239 had lost its dominance status since 2013 (Fig. 1A) (from 15.47% in 2013 to 3.35% in 2018 to 2020, $P < 0.0001$). Meanwhile, ST59-MRSA increased gradually in hospital visits (Fig. 1A) (from 3.31% in 2013 to 9.35% in 2018 to 2020, $P < 0.01$) . Interestingly, ST398-MRSA emerged within the ST398 clone. Further, a total of 81 *S. aureus* ST59 isolates and 43 *S. aureus* ST398 isolates were involved in phylogenetic construction separately (Fig. 1B and C), and no apparent geographic clustering was observed for either STs, suggesting that the phylogenetic structure could not be fully explained by geographic

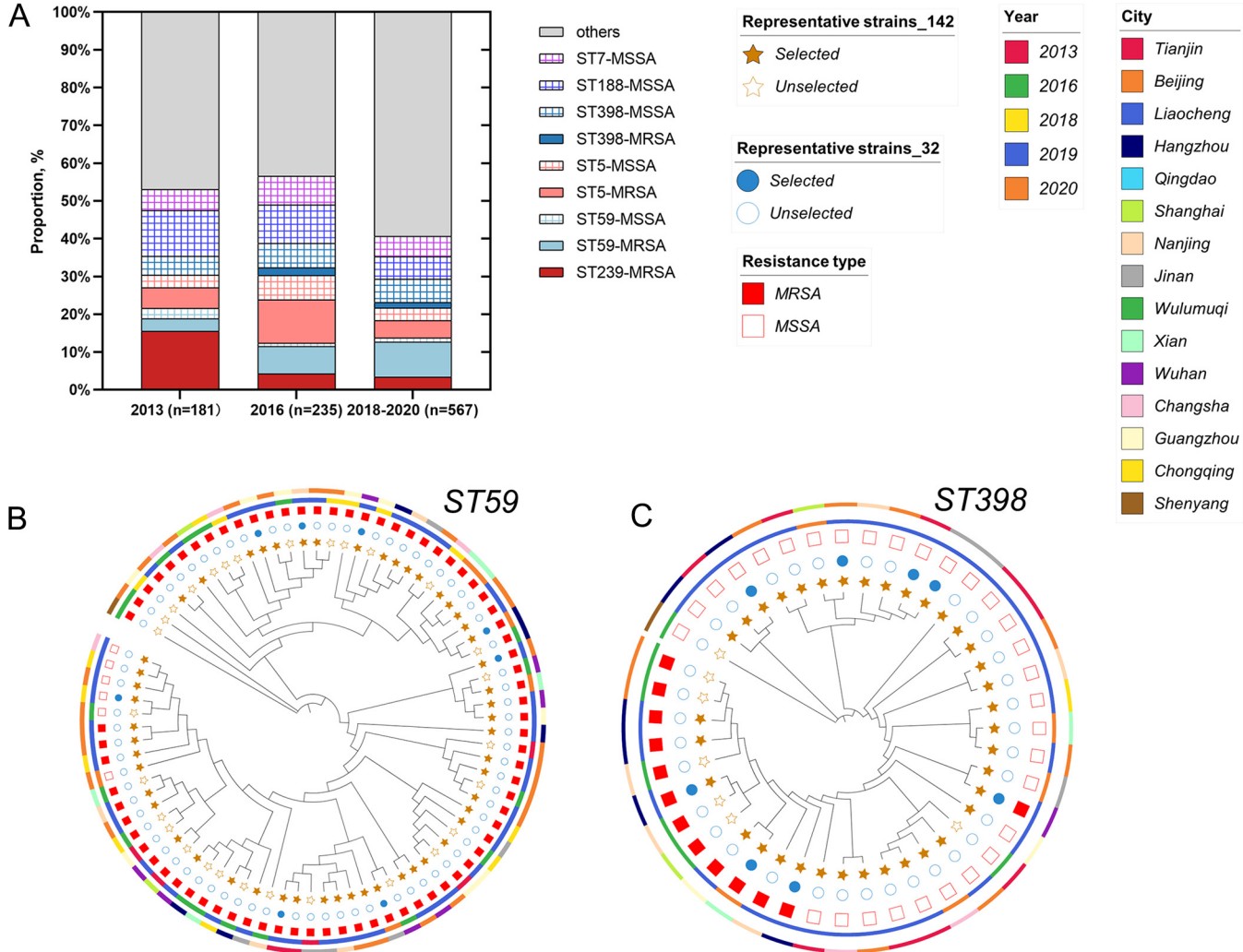

**FIG 1** Epidemic and phylogenetic features for ST59 and ST398. (A) The dynamic changes of epidemic *S. aureus* clones from bloodstream infections of inpatients in China from 2013 to 2020. (B, C) Phylogenetic analysis of the epidemic clones ST59 and ST398. The trees were constructed by IQ-TREE, based on the core gene alignment generated by Roary, annotated by the iTOL Web tool. Strain informations were mapped on the tree, from inner to outer circle: 1) The represented 142 strains selected in ST59 and ST398 were painted as filled yellow stars. 2) The represented 32 strains were painted as filled blue dots. 3) The filled red square represented the presence of *mecA*. 4) The circle colors distinguished the different collection years. 5) The circle colors distinguished the different locations.

sampling in either ST398 or ST59 clone, with isolates collected from hospitals located in different Chinese cities interspersed in the phylogeny. Furthermore, no collection-year clustering was observed. It was evident that ST398-MRSA and MSSA belonged to different lineages, suggesting that they owned unique genetic backgrounds.

**Lineage-specific genes among the 142 represented strains.** To illustrate the genomic characteristics of the epidemic *S. aureus* clones, a total of 142 recent isolates (2019 to 2020), including ST59 and ST398 (represented strains are painted as filled stars or blue dots on the inner ring in Fig. 1B and C) and their counterparts ST239 and ST5, were involved. The distribution of enzyme, cytolytic, and superantigen coding genes were compared (Fig. 2A). The serine protease genes (*spl* cluster) were present only in ST5 and ST239, and the hyaluronate lyase precursor gene (*hysA*$^{VSa\beta}$), protease synthase and sporulation protein gene (*paiB*) and CPBP family intramembrane metalloprotease gene (*cfim*) were specific to ST59 and ST398. For cytolytic genes, luk D/E were present only in ST5 and ST239, and *pvl* was seldom found in our strain collections (only in five ST59-MRSA, two ST398-MSSA, and one ST398-MRSA isolates). ST59 had a higher prevalence of *seb*, *selk*, and *selq*, while *sea* was prevalent in ST239. The ST5-MRSA clone possessed *sec* and *sell*. Interestingly, the *tsst-1* gene was found only in ST5-MRSA.

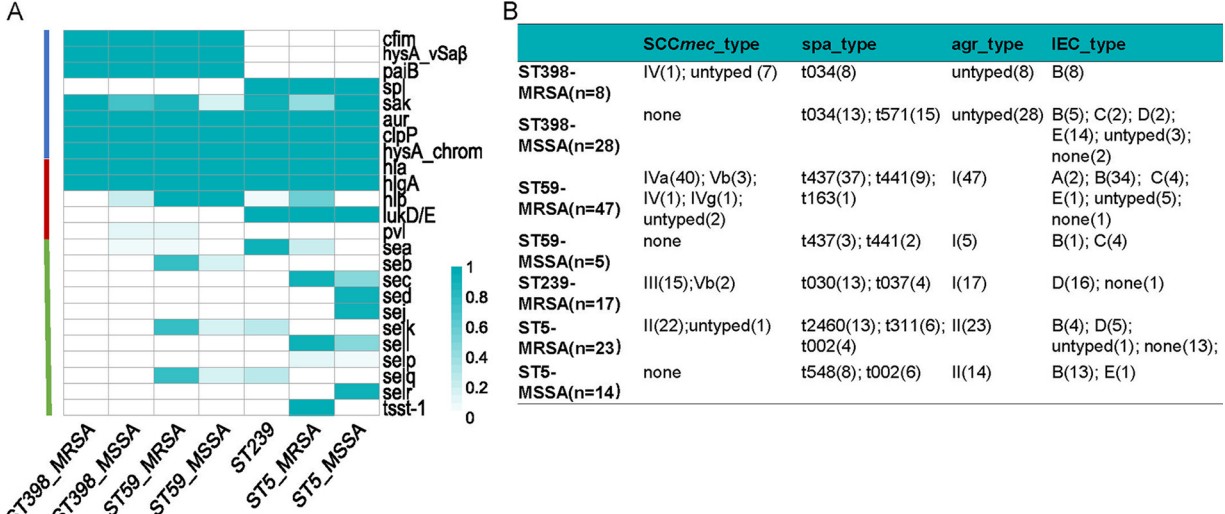

**FIG 2** Genetic traits among the represented 142 *S. aureus* strains. (A) Heat map of the enzyme, cytolytic, and superantigen-associated genes; each cell in the heat map indicates the percentage of the virulence gene in specific clones. The aureolysin (*aur*) ClpP protease (*clpP*), hyaluronidase (*hysA*, located on chromosome), *hla*, and *hlgA* were conserved among the different clones. (B) The SCC*mec* type, spa_type, agr_type, and immune evasion cluster (IEC) type for the 142 strains grouped by different resistance and clonal type combinations; the numbers in brackets represent the number of strains.

As displayed in Fig. 2B, the immune evasion cluster (IEC) type was identified for each clone (19). Most of the ST59 isolates belonged to type B (34/47, 0.723), while ST239 belonged to type D (16/17, 0.941). Interestingly, ST59, ST398-MRSA, and ST5-MSSA exhibited a similar pattern to IEC type B (see Table S3 in the supplemental material).

**Proteolysis-associated phenotypic and genetic environment analysis.** Proteolytic activity differed between different STs (Fig. 3A). Notably, ST398-MRSA showed the highest proteolytic ability on 2% skim milk-tryptic soy agar (TSA) plate. ST5-MRSA exhibited the lowest proteolytic ability compared with the other clones ($P < 0.0001$). No significant differences were observed between the ST59 and ST239 group, even though an increased trade was observed in ST59.

Genetic environment analysis suggested that ST59 lacks the typical *v*Sa$\beta$ structure because of the insertion of staphylococcal prophage $\phi$Sa3 however, it retained the hyaluronate lyase precursor gene (*hysA*$^{VSa\beta}$), which is shorter than the ST398 *hysA*$^{VSa\beta}$. In contrast, ST5-MRSA/MSSA and ST239 possessed the complete *v*Sa$\beta$ structure of an enterotoxin gene cluster (*EGC*), *spl* cluster and type I RM system (*hsdM* and *hsdS*). As shown in Fig. 3C, upstream of *paiB* is the cold shock protein-encoding gene *cspA* followed by the modulator of sarA (*msa*), which is involved in virulence gene expression and stress response. The 3′ end of *paiB* is a lysine biosynthesis operon. Further analysis of *paiB* suggested it belongs to the transcriptional regulator MerR family. Compared with ST239 and ST5, ST59 and ST398 had two copies of *cfim*. Whether these enzyme genes contribute to the enhanced proteolytic capacity of ST59 and ST398 remains to be determined.

**ST239-MRSA and ST5-MRSA showed impaired fitness while growing *in vitro*.** To further confirm the growth characteristics of different clones, one representative strain used for the growth assay was selected randomly from each clone. The growth rate varied significantly among the representative strains (Fig. 4A). We observed that ST239 showed impaired growth fitness compared with ST59-MRSA ($P < 0.0001$) or ST398-MRSA ($P < 0.0001$). Notably, ST5-MRSA grew slower in the tryptic soy broth (TSB) medium than ST5-MSSA ($P < 0.0001$).

**ST59 and ST398 had higher cytolytic potential *in vitro*.** The results obtained for hemolysis of the 142 strains were shown in Fig. 4B. ST59 and ST398 had a significantly higher hemolysis phenotype than ST239 ($P < 0.0001$). Interestingly, ST5-MSSA displayed a higher hemolytic ability than its counterpart ST5-MRSA ($P < 0.0001$). A similar pattern was

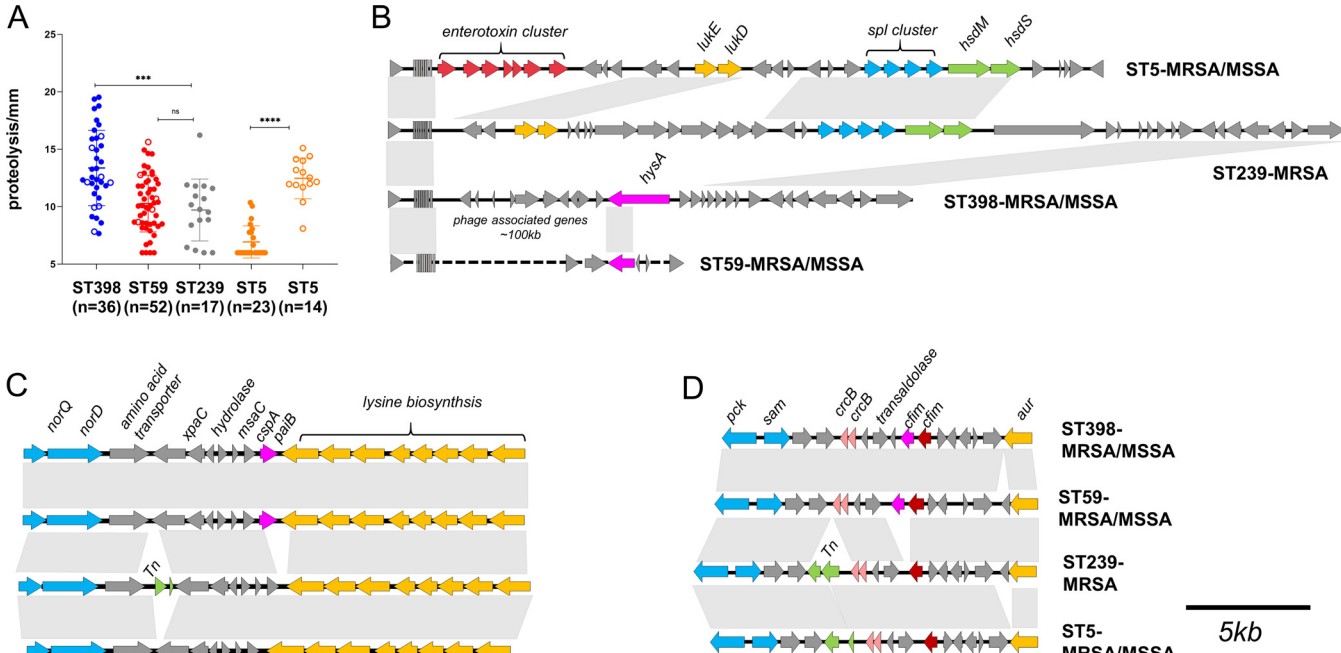

**FIG 3** Proteolysis-associated phenotypic and genetic environment analysis. (A) Proteolytic activity is assessed on TSA containing 2% skim milk among the different clones. (B) Genomic island νSaβ carrying the operon *spl* cluster that is complete in ST239 and ST5, *hysA*νSaβ is specifically presented in ST59 and ST398 with different sequences, and ST59 has a disrupted νSaβ structure with prophage-associated genes inserted. (C) Genomic region encompassing the gene *paiB*, which is missing in ST239 and ST5 genomes. (D) The CPBP family intramembrane metalloprotease gene (*cfim*) and surrounding genes among the four clones. Statistical significance of proteolysis abilities was determined by using the unpaired Student's *t* test. Data are shown as mean ± SEM. Each dot represents a methicillin-resistant (MRSA, filled dot) or methicillin-susceptible (MSSA, hollow circle) isolate. ****, $P < 0.0001$; ns, not significant ($P \geq 0.05$). Genes are indicated by arrowed boxes and colored based on gene function classification. The gray boxes with no function annotations represent hypothetical protein-encoding genes.

observed among the different STs for *RNAIII* and *psmα* expression levels (Fig. 4C and D), which indicates the virulence variations among different STs. Interestingly, while searching for hemolysin genes in the 142 strains, we found that the sphingomyelinase gene *hlb* remained intact in most of the ST59 (49/52) and some of the ST5-MRSA (13/23) strains, with higher expression than ST239, ST398, and ST5-MSSA (see Fig. S2 in the supplemental material). This finding might promote the cytolytic ability of ST59 to some extent.

**ST398 and ST59 were recognized strongly by human neutrophils.** The host-pathogen interaction is crucial for the pathogenesis of invasive bacterial infections. In order to further evaluate the role of ST59 and ST398 in causing immune responses while interacting with neutrophils, we analyzed the bacterial survival rate of different clones. The results indicated that ST239-MRSA and ST5-MRSA showed stronger resistance to polymorphonuclear leukocyte (PMN)-mediated killing (Fig. 5A) ($P < 0.05$) than ST398 and ST59. Furthermore, the annexin V/propidium iodide (PI) assay showed that ST398, ST59, and ST5-MSSA caused more neutrophil apoptosis than ST239-MRSA and ST5-MRSA (Fig. 5B). Taken together, our bacteria-neutrophil interaction experiments strongly suggested that unlike ST239-MRSA and ST5-MRSA, ST59 and ST398 appeared to be potent proinflammatory microbial invaders to the host cells. However, as mentioned above, we observed growth defects in ST239 and ST5; thus, the long-term survival rate in the host may not be achieved.

**NETs formation and degradation were pronounced in ST59.** NETs formation and degradation assays were conducted on representative isolates. As shown in the image (Fig. 6C), after 2 h of incubation, we observed that ST59 and ST5-MSSA showed remarkable NETs degradation in the DNA/MPO strain and merged image, while ST239 and ST5-MRSA were scarcely degraded. The proportion of NETs per total number of neutrophils was calculated to compare NETs formation in five individual images per sample for different bacterial strains. ST59 displayed increased NETs formation ability

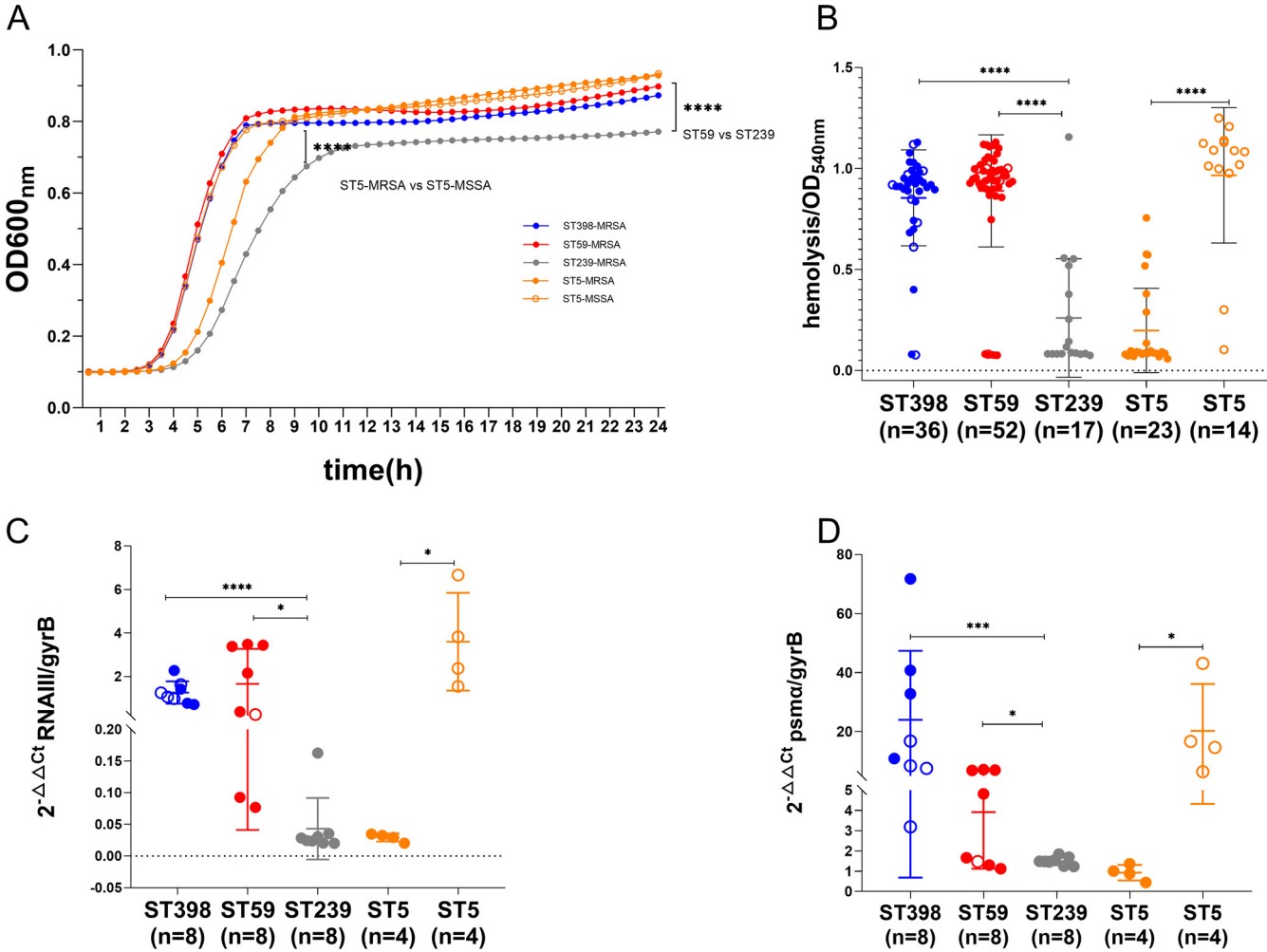

**FIG 4** *In vitro* cytolytic capacity comparison. (A) *In vitro* growth curve among the different STs; ST239-MRSA and ST5-MRSA showed impaired fitness during growth *in vitro*. (B) Toxicity for each isolate was determined by incubating the bacterial supernatant with human red blood cells (RBCs), using absorbance at 540 nm to quantify hemolysis ability relative to the hypervirulent strain 8325. (C, D) Eight strains from each clone were selected to compare the *RNAIII* (C) and *psmα* (D) expression levels with qPCR. The results were normalized to *gyrB* expression with RN4220 as the negative control. The Wilcoxon signed-rank test was used to compare differences of the growth curve. Statistical significance of cytolytic abilities and gene expression was determined by using the unpaired Student's *t* test and Mann-Whitney U test. Data are shown as mean ± SEM. Each dot represents a methicillin-resistant (MRSA, filled dot) or methicillin-susceptible (MSSA, hollow circle) isolate. *, $P < 0.05$; ***, $P < 0.001$; ****, $P < 0.0001$.

compared with ST239 (Fig. 6B) ($P < 0.0001$), and there was no significant difference between ST5-MRSA and ST5-MSSA. The cell counting kit-8 (CCK8) assay was used to compare NAD$^+$ release ability after bacterial infection. Interestingly, we found that neutrophils were highly active when exposed to ST59, ST398, and ST5-MSSA (Fig. 6A) (ST59 versus ST239: $P < 0.0001$). It was suggested that ST398, ST59, and ST5-MSSA are stronger stimulations to neutrophils, reflected by an increased NAD$^+$ release and NETs formation and degradation.

**Adhesion-associated genotypic and phenotypic study of the different STs.** Except for *in vitro* virulence comparisons, *S. aureus* fitness contributes to transmission and clonal success (2). Variation was found in genes encoding cell surface-bound proteins among different clones (Fig. 7A), such as the collagen adhesin gene *cna* (absent in ST59 and ST5, deletion of B domain in ST239 and ST398-MSSA). ST59 was decay of *sasD*, and ST59 and ST5 harbored domain deletions in the extracellular adhesion gene *eap*. ST239 lacked the fibronectin-binding protein B-coding gene *fnbB* (in 82.3% of isolates) and had a premature von Willebrand factor-binding gene, *vwb*. The serine aspartate repeat gene D/E (*sdrD/E*) was absent in ST398-MRSA and parts of ST398-MSSA (17.8% and 10.7%, respectively). However, the other adhesion-associated genes *clfA/B*,

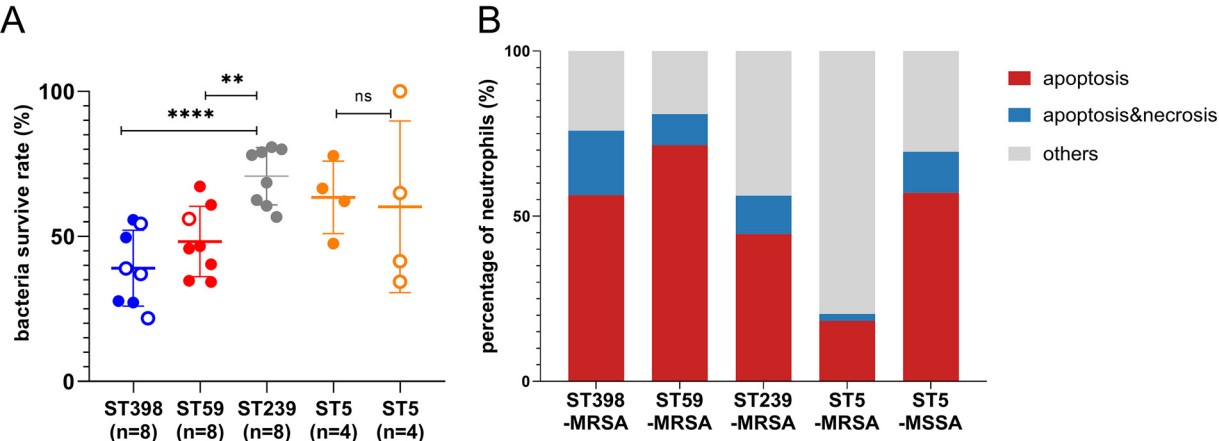

**FIG 5** ST59 and ST398 are more potent microbial invaders when confronted with neutrophils. (A) Bacterial survival rate of different clones after incubating with freshly isolated neutrophils for 30 min. (B) Apoptosis and necrosis of neutrophils were confirmed by annexin V-propidium iodide (PI) staining and flow cytometry analysis; bacterial supernatants were 10-fold diluted. Statistical significance of the bacterial survival rate was determined by using the unpaired Student's *t* test. Data are shown as mean ± SEM. Each dot represents a methicillin-resistant (MRSA, filled dot) or methicillin-susceptible (MSSA, hollow circle) isolate. **, $P < 0.01$; ****, $P < 0.0001$; ns, not significant ($P \geq 0.05$).

*ebps*, *fnbA*, *isaB*, *isdA/B*, *sasC*, *sasF*, *sdrC/E*, and *emp* (data not shown) remained conserved in our collection of isolates.

Among the contributors to the fitness of *S. aureus* is a range of transmembrane transporters that can mediate the uptake of nutrients or export of metabolites and toxic substances, of which the oligopeptide ABC transporter system (Opp system) has been shown to promote adhesion to human epithelial cells and nutrient transport functions (20). The *S. aureus* strains in our collection encoded five putative peptide transport systems (see Table S2 in the supplemental material). Notably, ST398 carried the Opp-3′ operon (OppAFDBC) located 212 bp downstream of the Opp-3 operon, while the other clones harbored the Opp-4 operon (Fig. 7B). The OppA-3′ in ST398 displayed 70.2% sequence similarity with OppA-3. Furthermore, the cell adhesion and invasion assays were performed in 32 representative strains from the 4 clones using the human epidermal keratinocytes (HaCaTs). We revealed that ST398 showed higher adhesion and invasion abilities than other STs (Fig. 7C) ($P < 0.05$).

## DISCUSSION

In this study, we discovered lineage-specific traits for the epidemic *S. aureus* clones ST59 and ST398 compared with ST239 and ST5. ST59, ST398, and ST5-MSSA showed a higher expression of RNAIII and *psmα*, as well as proficiency at causing cell lysis (Fig. 4) and proteolysis (Fig. 3). We revealed lineage-associated genetic traits, including *paiB*, *cfim*, *hysA^{VSaβ}*, and *hlb*, in the epidemic clones (Fig. 2) that might contribute to their higher virulence; functional verification is under way. ST59 and ST398 were strongly recognized by human neutrophils and caused more cell apoptosis and NETs degradation, suggesting their potential in host-pathogen interactions (Fig. 5 and 6). Additionally, these strains differed substantially in their repertoire and composition of intact adhesion genes. ST398 displayed higher adaptability to human epidermal keratinocytes and a unique genetic arrangement inside the Opp system, indicating functional variations (Fig. 7 and Fig. S2).

To our knowledge, this is the first study to compare differences at the genomic and host cell-pathogen interaction phenotypic levels for the four MRSA clones. Our results differed from recent observations reported by our previous research (13) in some respects, as follows: (i) more clones (ST59, ST398, ST239, and ST5) involved in this study; (ii) we compared the host cell-pathogen interactions, especially for neutrophils and keratinocytes; and (iii) we reported lineage-specific genetic traits that had not been reported before, such as *paiB*, *cfim*, and *cna*.

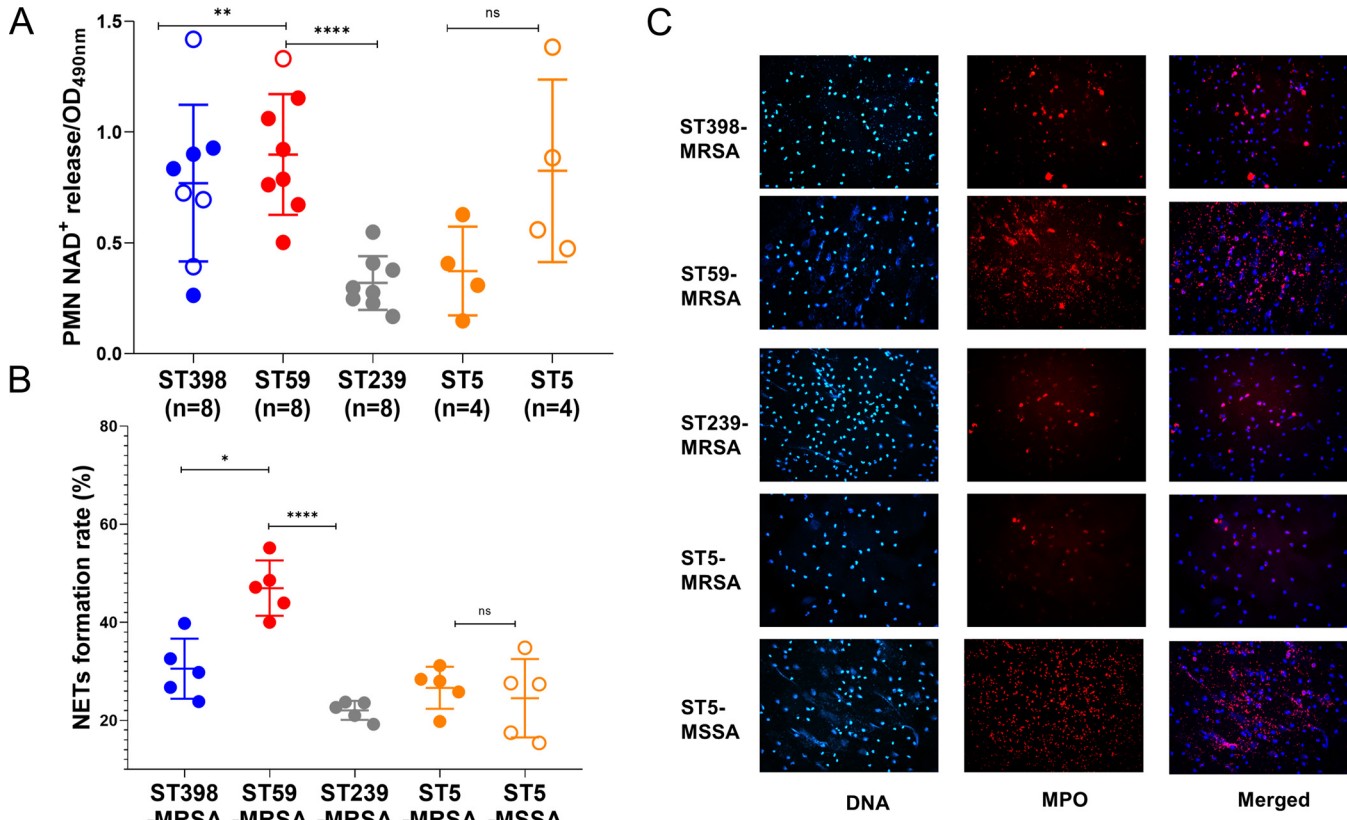

**FIG 6** NETs formation and degradation abilities among the represented strains. (A) The CCK8 assay was used to compare the neutrophil NAD$^+$ release ability after bacterial infection; eight representative strains are presented from each clone. (B) The proportion of NETs per total amount of neutrophils was calculated to compare NETs formation in five individual images per sample. (C) Representative picture for NETs formation for one isolate from each clone, visualized by DNA and MPO stain. Statistical significance of NAD$^+$ release and NETs formation rate were determined by using the unpaired Student's *t* test. Data are shown as mean ± SEM. Each dot represents a methicillin-resistant (MRSA, filled dot) or methicillin-susceptible (MSSA, hollow circle) isolate. *, $P < 0.05$; **, $P < 0.01$; ****, $P < 0.0001$; ns, not significant ($P \geq 0.05$).

The gain and loss of virulence determinants carried on MGEs play vital roles in bacterial adaptability, virulence, and survival (21). Our study suggested that the genomic island *vSaβ* displayed lineage-associated variations, *hysA$^{vSaβ}$* specifically presented in ST59 and ST398 with sequences variations (Fig. 3), and *hysA$^{vSaβ}$* was also reported to be present in LA-MRSA ST398 (22), suggesting it may have virulence potential. Consistent with the previous findings (7), the *S. aureus* bacteriophage *ϕ*Sa3 atypically integrated into ST59 in our analysis without disrupting the integrity of *hlb*, an important virulence factor in skin colonization and chronic inflammatory diseases (23, 24). Thus, one of the most interesting features of ST59 is its higher expression level for *hlb* than the other STs (Fig. S2), indicating its role in increasing pathogenicity in ST59. Collectively, the different compositions of MGEs within the four STs suggest that different strategies have been adopted by different STs to successfully overcome host response and cause infection.

Virulence and antibiotic resistance have played essential roles in shaping the success of epidemic clones in hospital settings. Proteases play a role in nutrient acquisition, bacterial dissemination, and immune evasion (12). A genome-level comparison identified different patterns for protease-associated genes among the four clones. ST239 and ST5 harbored the *spl* cluster, while *paiB*, *cfim*, and *hysA$^{vSaβ}$* have a higher prevalence in ST59 and ST398. The phenotypic comparison suggested that ST59 and ST398 had an enhanced proteolysis capacity (Fig. 3). Whether these lineage-specific genes might play a role in the proteolytic phenotype remains to be determined. Consistent with previous studies (7–9), we showed that ST59 and ST398 had fewer antibiotic resistance genes and were susceptible to most antibiotics compared with ST239

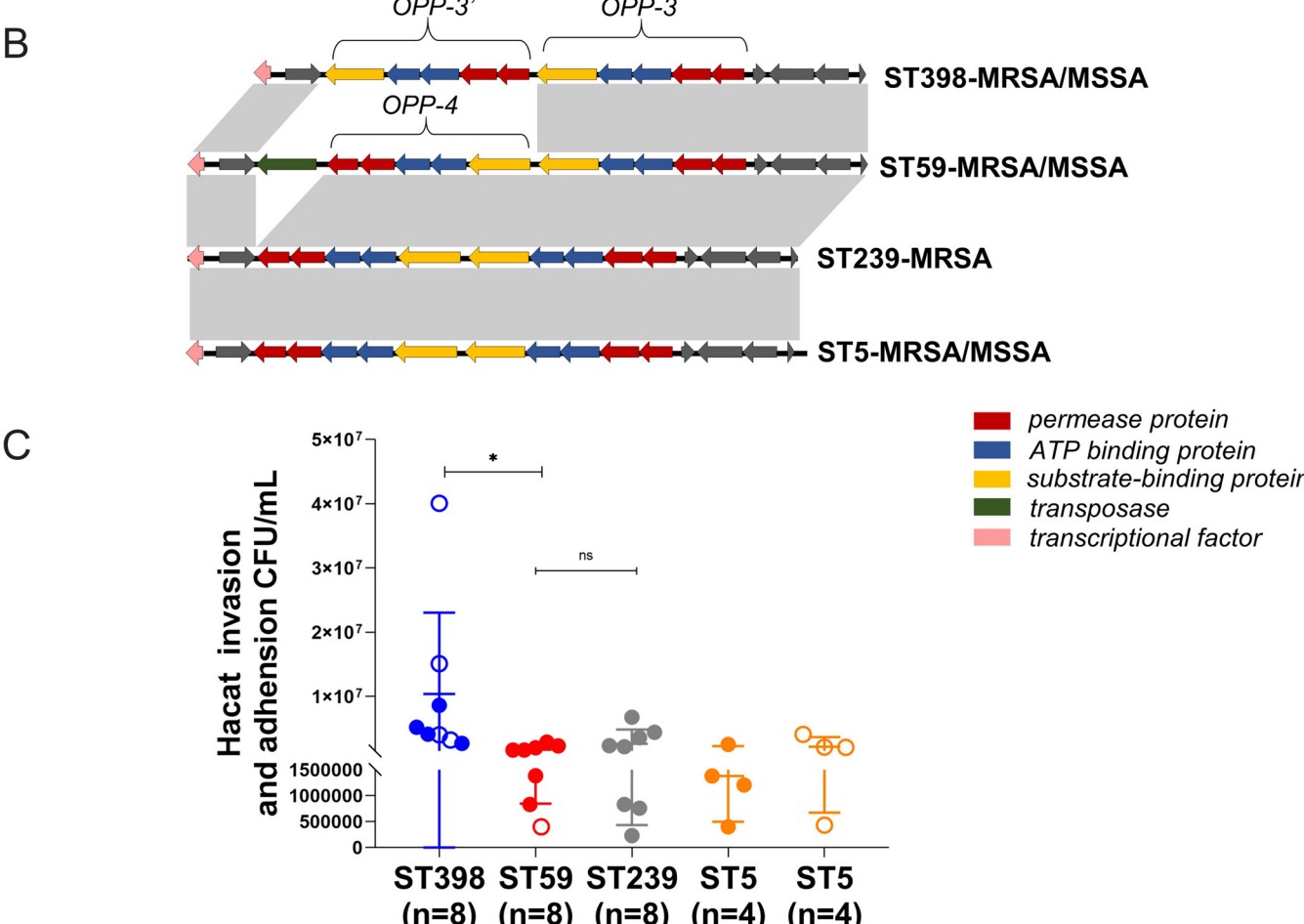

**FIG 7** ST398 has unique Opp system, adhesion-associated genes, and higher adhesion and invasion capacity with keratinocytes. (A) Variation of the surface protein-encoding genes among the different clones in 142 isolates (*cna* in 32 strains, because of the analysis of *cna* depends on nanopore sequencing results) * refers to the percentage of gene absent. (B) Different structures within the Opp system between ST398 and the other clones, compared by the Easyfig tool. Genes are indicated by arrowed boxes and colored based on the Opp system gene function classification. The gray boxes with no function annotations represent hypothetical protein-encoding genes. (C) Cell adhesion and invasion capacity in keratinocytes among the 32 strains, separated by different clones. Statistical significance of adhesion and invasion abilities was determined by using the Mann-Whitney U test. Data are shown as mean ± SEM. Each dot represents a methicillin-resistant (MRSA, filled dot) or methicillin-susceptible (MSSA, hollow circle) isolate. *, $P < 0.05$; ns, not significant ($P \geq 0.05$).

and ST5-MRSA (see Fig. S3 in the supplemental material). Antibiotic resistance can cause *agr* system dysfunction and increase fitness costs to *S. aureus* (25, 26). According to our analysis, -MRSA and ST5-MRSA had a reduced cytolytic ability and *in vitro* growth rate (Fig. 4). In contrast, the *agr* system remained active in ST59 and ST398 as important virulence determinants, which is comparable to the previous study (14).

In particular, we discovered that ST5-MSSA had a higher virulence potential than ST5-MRSA, consistent with previous findings (27). Further analysis revealed a higher expression of *psmα* and *RNAIII* in ST5-MSSA (Fig. 4). This tendency was not observed for ST59-MRSA/MSSA or ST398-MRSA/MSSA, which displayed active *agr* functions. However, the mechanism that causes ST5-MSSA to become highly virulent remains elusive.

In addition to *in vitro* virulence factors, such as proteolysis and cytolytic abilities, host-pathogen interactions are also important for successful epidemic *S. aureus* clones. Neutrophils are usually the first cells recruited to the infection site, where they are activated and kill bacteria by an arsenal of antimicrobial processes, such as reactive oxygen species (ROS), antimicrobial granules, and NETosis (18). Our results indicate that the hypervirulent clones ST59 and ST398 were effective microbial invaders that cause more neutrophil apoptosis and that they were eliminated faster by the host immune cells, such as neutrophils, than ST239-MRSA and ST5-MRSA (Fig. 5). The same trend was observed in fungal infections, where *Candida albicans* appeared to be less immunoinflammatory and caused more cell death than *Candida auris* (28). When comparing the NETosis ability of different STs, we discovered that ST59 and ST398 promoted NETs formation and showed higher NADPH oxidase activity than ST239-MRSA and ST5-MRSA (Fig. 6). As the release of NETs is linked with immune production, increased NETs formation by ST59 could cause severe bystander effects on the host by inflammation and tissue damage. Notably, we discovered that NETs degradation was pronounced in ST59 and ST5-MSSA, suggesting that they could easily destroy released NETs, interact with microbicidal components, and circumvent NETs' microbicidal properties (29). In contrast, no damaged NETs were observed after ST239-MRSA and ST5-MRSA stimulation (Fig. 6).

Our previous study confirmed the high prevalence of ST398 associated with SSTIs (30). In the present study, we provide evidence that ST398 was superior to other clones in its ability to adhere and invade human skin epidermal keratinocytes. We revealed the presence of variations and absence of major domains in genes encoding surface adhesion proteins, such as *cna*, *eap*, and *fnbB*, among different clones. This finding suggests a differential adaptation to host extracellular matrix proteins among different clones. The *sdrD/E* gene decay in ST398-MRSA and the small percentage of ST398-MSSA may give priority to ST398-MSSA to adhere to human cells (Fig. 7). Moreover, the Opp system has been shown to contribute to adhesion to human epithelial cells (20), and ST398 was found to have a different arrangement pattern than the other clones (Fig. 7 and Fig. S3). We propose that this result is likely a reflection of genome duplication events (31). The importance of these determinates in adhering to human epidermal keratinocytes for ST398 strains remains to be determined. What confused us was the low adhesion and invasion capacity of ST59 in our study since it has been recognized as the main CA-MRSA in China, whereas in another study, it was claimed that the *pvl*-positive ST59 clone has an advantage in adhesion over other clones of MRSA (32). This difference may be attributed to the fact that most ST59 strains lacked *pvl* in our collection. Further *in vivo* assays are required to clarify this phenomenon.

In summary, the success of epidemic *S. aureus* clones is multifactorial, with phenotypic and genomic traits involved. As we broaden our knowledge of specific genomic traits coupled with *in vivo* host-pathogen interactions, we will be able to portray a comprehensive overview of the clonal replacement phenomenon. However, the limitation of this study is the lack of an *in vivo* virulence comparison among the different STs and the functional validation of lineage-specific genes. This understanding will enable the development of intervention strategies for the most effective targets to treat clinical *S. aureus* infections.

## MATERIALS AND METHODS

**Bacterial isolates.** We performed spa typing on 983 strains collected from our 7-year national survey of *S. aureus* BSI (*n* = 983) and multilocus sequence type (MLST) analysis on 833 strains of the epidemic spa type (which can represent dominant STs). The other 150 strains were classified into the other ST

groups, and a total of 142 recent (2019 to 2020) represented isolates, including MRSA and MSSA, were selected for further analysis. The isolates were identified using standard techniques. The strain selection procedure is illustrated in Fig. S1 in the supplemental material.

All patient data were anonymized. Ethical approval and informed consent were not required. Ethical approval and consent to participate in bacterial strains were collected in routine clinical diagnostics, and ethical approval and consent to participate from patients were not required. The analysis of deidentified patient demographic data was approved by the Peking University People's Hospital Institutional Review Board (no. 2020PHB193-01). The study conformed to the principles of the Declaration of Helsinki and its later amendments.

**Whole-genome sequencing, phylogenic, and molecular analysis.** A total of 81 ST59 and 43 ST398 isolates and the represented *S. aureus* strains of other clones were subjected to whole-genome sequencing on Illumina, and the selected 32 strains were sequenced on Nanopore platforms. *De novo* assembly was performed by SPAdes, and hybrid assembly was performed using Unicycler v0.4.7. The assembly was annotated using Prokka 1.13.7 and the online service Rapid Annotation using Subsystem Technology (RAST; http://rast.nmpdr.org/).

The core genome of *S. aureus* was determined using the pangenome analysis pipeline Roary v3.12.2 (33). Maximum likelihood phylogenetic trees were constructed using IQ-TREE software (34). Finally, a tree was plotted and annotated using the iTOL Web tool.

MLSTs were assigned using PubMLST (https://pubmlst.org/saureus/). spaTyper 1.0 was used for spa typing, and SCC*mec*Finder 1.2 (default threshold, 90% identity; 60% minimum length) was used for SCC*mec* typing. Antibiotic resistance genes were detected using Resfinder (https://cge.food.dtu.dk/services/ResFinder/) and the Comprehensive Antibiotic Resistance Database (CARD; https://card.mcmaster.ca). Virulence and adhesion-associated genes were characterized using blastn research. Genomic islands and genomic regions of interest were identified using blastn and further compared using the Easyfig tool (http://mjsull.github.io/Easyfig/).

**Proteolysis assay.** The proteolytic activity of the 142 isolates was assessed using TSA containing 2% skim milk. Overnight cultures were placed on the skim milk agar plates and incubated at 37°C for 21 h. Lysis halos were measured from the border of the spotted bacteria to the edge of the clearance zone. Each strain was analyzed in at least three biological replicates. Statistical significance of proteolysis abilities was determined by using the unpaired Student's *t* test.

***In vitro* cytolytic assay.** Culture filtrates were used to lyse erythrocytes. A total of 142 *S. aureus* strains were cultured for 18 h in TSB, and culture supernatants were added to human red blood cells (2% [vol/vol] in phosphate-buffered saline [PBS]) and incubated at 37°C for 1 h with gentle shaking. Cells with PBS or TSB were used as blank controls. The culture was then centrifuged at $1,500 \times g$ for 10 min at 4°C without disturbing the cells. The supernatant was transferred to a new 96-well plate, and hemolysis was determined by measuring the optical density at 540 nm using a spectrophotometer. The assay was performed in triplicate. The statistical significance of cytolytic abilities was determined by using the unpaired Student's *t* test.

**Quantitative reverse transcriptase PCR (qRT-PCR).** For quantitative PCR (qPCR) analysis, MRSA strains were grown in TSB to mid-log phase, and their RNA was isolated using the RNeasy minikit (Qiagen, 74104) and transcribed to cDNA (Qiagen, RR036A), according to the manufacturer's instructions. qPCR was performed by amplifying 20 ng of cDNA in a 20 $\mu$L total reaction volume with TB green premix Taq II on the 7500 detection system (ABI, USA) under the following conditions: 30 s at 95°C, 40 cycles of 3 s at 95°C, and 30 s at 60°C, followed by a dissociation curve. No-template controls were performed in parallel. The primers used for qPCR are listed in Table S1 in the supplemental material. Data were analyzed, and relative expression (cycle threshold [$\Delta C_T$]) was determined by using ABI 7500 software v2.3. The expression was normalized to that of gyrB, with each value representing three biological replicates. The statistical significance of gene expression was determined by using the unpaired Student's *t* test and Mann-Whitney U test.

**Human PMN assay. (i) Neutrophil bactericidal assay.** The density gradient separation method was used to isolate healthy human neutrophils from whole blood as described previously (35). Erythrocytes were sedimented using dextran, lysed with ACK lysing buffer, and suspended in phenol red-free RPMI 1640 at the final concentration. The human PMN bactericidal assay was performed as described previously (36). Briefly, bacteria in the exponential growth stage were diluted to $10^7$ CFU/mL with phenol red-free RPMI 1640. Human neutrophils (100 $\mu$L, $1 \times 10^6$ cell/mL) were mixed with opsonized bacteria (20 min; 10% serum) in 1.5-mL centrifuge tubes, centrifuged at 1,500 rpm for 5 min at 4°C, and cultured at 37°C and 5% $CO_2$ for 30 min. PMNs were dissolved immediately in 0.2% Triton X-100 (Sigma-Aldrich, X100), serially diluted in $1\times$ PBS, and plated onto TSA plates. The percentage of *S. aureus* survival in PMNs was calculated the following day, using the equation ($CFU_{PMN+}$ at t/$CFU_{PMN-}$ at t0) $\times$ 100. The statistical significance of the bacterial survival rate was determined by using the unpaired Student's *t* test.

**(ii) Apoptosis and necrosis detection.** The apoptosis of neutrophils was confirmed by annexin V-propidium iodide (PI) staining flow cytometry analysis. The overnight bacterial supernatant was diluted 10-fold and incubated with 50 $\mu$L freshly isolated neutrophils ($1 \times 10^6$ cell/mL) for 30 min at 37°C under static conditions. Cell apoptosis and necrosis were quantified according to the manufacturer's instructions (Beyotime, C1062S).

**NETs formation and stain.** The neutrophil suspension (50 $\mu$L) was seeded into 96-well plates, and logarithmic growth period bacteria were washed by phenol red-free RPMI 1640 twice and infected with neutrophils at a multiplicity of infection (MOI) of 100:1. The mixture was incubated at 37°C for 2 h. Next, the cell counting kit-8 (CCK8) reagent was added and incubated at 37°C for 2 h, and optical density at 450 nm ($OD_{450}$) was measured to determine $NAD^+$ release. The statistical significance of $NAD^+$ release was determined by using the unpaired Student's *t* test.

NETs induction, staining, and visualization were performed as described previously (35). Isolated neutrophils were seeded on poly-L-lysine-coated glass coverslips in 24-well plates and allowed to settle

on the coverslip for 0.5 h. Phorbol myristate acetate (PMA) was added to stimulate neutrophils to release NETs as a positive control. Bacteria were applied at an MOI of 10:1. After 2 h, cells were fixed with paraformaldehyde overnight at 4°C. Fixed neutrophils were blocked with blocking buffer (0.2% gelatin in PBS) at 37°C for 1 h. In a humid chamber, cells were stained with anti-human MPO primary antibody at 37°C for 2 h. After three washes with PBS, the cells were incubated for 1 h with a donkey anti-rabbit IgG secondary antibody conjugated with Alexa Fluor 555. DNA was stained with Hoechst 33342. NETs were visualized using fluorescence microscopy. The statistical significance of NETs formation rate was determined by using the unpaired Student's *t* test.

**Adhesion and invasion of *S. aureus* to human skin epithelial cells.** Cell adhesion and invasion assays were conducted as described before (27). HaCaTs were cultured in Dulbecco's modified Eagle's medium (DMEM; high glucose) with fetal bovine serum (FBS; 10%) at 37°C and 5% $CO_2$. Bacterial pellets were suspended in DMEM and added to monolayers (>90% confluence) at an MOI of 10 in 24-well plates, followed by incubation for 2 h. To remove nonadherent bacteria, the supernatants were discarded, and the cells were washed three times with sterile PBS. Cells were subsequently lysed using 0.2% TritonX-100, and bacterial CFU was enumerated by proper dilution of epithelial cell lysates and plated onto TSA plates. The assay was performed in triplicates. The statistical significance of adhesion and invasion abilities was determined by using the Mann-Whitney U test.

**Statistical methods.** GraphPad Prism 8 was used for statistical analysis. Student's *t* test, one-way analysis of variance (ANOVA), chi-square test or Fisher exact test, and Wilcoxon signed-rank test were performed to analyze statistical significance. For nonparametric data, the Mann-Whitney U test was applied. Error bars in all graphs indicated the standard deviation (mean ± SD). A *P* value of <0.05 was reported as statistically significant.

**Data availability.** Raw Illumina short-read data from the 157 newly generated *S. aureus* isolates are available on NGDC with BioProject identifier (ID) PRJCA012518 (https://ngdc.cncb.ac.cn/). Full accompanying metadata information are available in Data Set S1 in the supplemental material.

## SUPPLEMENTAL MATERIAL

Supplemental material is available online only.

**DATA SET S1**, XLSX file, 0.04 MB.
**FIG S1**, TIF file, 0.04 MB.
**FIG S2**, TIF file, 0.1 MB.
**FIG S3**, TIF file, 0.4 MB.
**TABLE S1**, DOCX file, 0.02 MB.
**TABLE S2**, DOCX file, 0.02 MB.
**TABLE S3**, DOCX file, 0.01 MB.

## ACKNOWLEDGMENTS

This work was supported by a National Natural Science Foundation of China (NSFC) grant (81991533).

H.W. conceived and designed the study. F.C., Y.Y., L.J., S.L., Q.W., and S.S. collected the data and performed experiments described in this study. Bioinformatic analyses were performed by F.C., H.C., R.W., and S.W. F.C. wrote the draft and H.W. revised it. The authors read and approved the final manuscript.

We declare that we have no competing interests.

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
