## [Reviewer comments · mSystems]

Phenotypic and Genomic Comparison of *Staphylococcus aureus* Highlights Virulence and Host Adaptation Favoring the Success of Epidemic Clones

Fengning Chen, Yuyao Yin, Hongbin Chen, Longyang Jin, Shuguang Li, Ruobing Wang, Shuyi Wang, Qi Wang, Shijun Sun, and Hui Wang

Corresponding Author(s): Hui Wang, Department of Clinical Laboratory, Peking University People's Hospital, Beijing, China

Review Timeline:

Submission Date:	August 31, 2022
Editorial Decision:	October 7, 2022
Revision Received:	October 19, 2022
Accepted:	October 27, 2022

Editor: Zackery Bulman

Reviewer(s): Disclosure of reviewer identity is with reference to reviewer comments included in decision letter(s). The following individuals involved in review of your submission have agreed to reveal their identity: Youjun Feng (Reviewer #1); Karine Dufresne (Reviewer #2)

Transaction Report:

DOI: <https://doi.org/10.1128/msystems.00831-22>

October 7, 2022

Prof. Hui Wang
Peking University People's Hospital
Department of Clinical Laboratory
No.11 Xizhimen South Street
Beijing 100044
China

Re: mSystems00831-22 (Phenotypic and Genomic Comparison of Staphylococcus aureus Highlights Virulence and Host Adaptation Favoring the Success of Epidemic Clones)

Dear Prof. Hui Wang:

Thank you for submitting your manuscript to mSystems. We have completed our review and I am pleased to inform you that, in principle, we expect to accept it for publication in mSystems. However, acceptance will not be final until you have adequately addressed the reviewer comments. Please see the specific reviewer comments below.

Preparing Revision Guidelines

Sincerely,

Zackery Bulman

Editor, mSystems

Journals Department
American Society for Microbiology
1752 N St., NW

Reviewer comments:

Reviewer #1 (Comments for the Author):

The manuscript by Chen et. al., "Phenotypic and Genomic Comparison of Staphylococcus aureus Highlights Virulence and Host Adaptation Favoring the Success of Epidemic Clones" described the genomic comparison of epidemic clones by genomic analysis. In addition, the author also confirmed the high virulence of ST59, ST398 and ST5-MSSA using vitro assays. Overall, I am impressed by the organization of number of new information regarding S. aureus. I also have couples of concerns, esp., no dominant clone identified in the MSSA strains. For clinical MSSA strains, several clones including ST5, ST398, ST188, ST88, ST7 and others comprise most of them, however, the ST59 clone which is identified as the dominant MRSA clone in China, accounted for not such a big proportion in MSSA. The MSSA lineages show considerable genetic diversity compared to the MRSA. For MRSA strains, there are indeed dominant clones. Moreover, the proportion of dominant clones in MRSA is much higher than that of other clones, and there are different dominant clones in different regions. Therefore, the authors should focus the main topic of this study on MRSA strains, due to there is no definition about the dominant clone in MSSA to date.

Abstract :

- 1.Line 34-36 : please delete the content of MSSA, and the enhanced proteolytic ability needs to be verified by experiments.
- 2.Line 39-40 : please re-write "Besides differences in virulence".
- 3.Line 51-52: there are no dominant clones in MSSA strains.
- 4.Do an English revision

Introduction

- 1.Line 65-67: This sentence is mainly aimed at MRSA and does not apply in MSSA. The object of this study should be narrowed down to MRSA, not the whole S.aureus group.
- 2.Line 70-71: ST398 has only been considered as CA-MRSA in the recent years, which is mainly based on related clinical data. In fact, most ST398 strains are considered as LA-MRSA. Misconception.
- 3.Line 81-82: The chp gene mainly participates in promoting immune evasion, i.e., helping bacteria escape from immune recognition, thus persisting in the host. Why does the chp gene enhance the cytolytic ability of ST59 clones?
- 4.please re-write line 82-85 to enhance its connection with the context.

Results

- 1.Line 102-116: : Which data is correct: 2013-2022 or 2020 ? Correct in all the text.
- Line 114 : the term "community clone" should be defined by clinical data rather ST types.
- 2.Line 119 : without phylogeographic and epidemiological data to support these statements, I don't think the present Figure give us any information to support this conclusion,
- 3.Line 128: Reasons for inclusion of ST5?
4. Line133: pvl was seldom found in ST59-MRSA and ST398-MSSA. I don't understand the meaning of this sentence. Do you want to express that the pvl gene exists in ST59 and ST398-MSSA, but it is not common in ST59 and ST398?
5. Line 136-137 : Interestingly, ST59, ST398-MRSA, and ST5-MSSA exhibited a similar pattern to IEC type B. What is similar pattern ?
6. Line 141-145: The paragraph is not clear, it is not understood. This paragraph is confusing. Since we are exploring the difference between dominant clones and non-dominant clones. Why do you compare ST398 with ST59, which is also a dominant clone. Why do you compare ST5-MRSA with ST5-MSSA? Please reselect clones to compare.
7. Line 174-176: The results indicate that ST239- and ST5-MRSA showed stronger resistance to PMN (Polymorphonuclear leukocytes) mediated killing. If ST239 and ST5 had stronger resistance to neutrophils, ST239 and ST5 should have stronger capacity to survive in the host The immune evasion ability is stronger, and it is not easy to be killed by immune cells. It means the stronger ability of ST239 and ST5 to persist in the host. However, the author stated that ST239 and ST5 are not the dominant clones of BSIs. How do the authors explain this results?
8. Line 185-196: Again, why compared ST5-MSSA with ST5-MRSA. The results shows that ST5-MSSA is the dominant clone of MSSA and it is more virulent. And ST5-MRSA is not a dominant clone, so if ST5-MRSA is weak in virulence. However, ST5-MRSA has a very high prevalence in Shanghai and Zhejiang provinces. If the high virulence account for the prevalence of the ST5-MSSA, what account for the ST5-MRSA in east China? How does the author explain it?
9. Line 217-220 : Could you describe a little more for details on this part of results ? The title is ST398 shows higher adhesion and invasion capacity with keratinocytes. But there is very little in this section.

Discussion

Line 235: paiB, cfim, hysAVSa β , and hlb replace "might contribute to ..." with "might be associated with ...". Do these genes really account for the success of epidemic clone? Actually we don't know.

Line 238: deleted ' . "

Line 242-244: Again, MSSA does not have dominant or epidemic clones. The definition of dominant and epidemic clones is

limited to MRSA.

M&M

Line 331-332: How the author selected the strains?

Reviewer #2 (Comments for the Author):

Chen et al studied the genetic and virulence phenotypes in between previously S. aureus epidemic lineages ST5 and ST239 compared to the now successful ST59 and ST398. Among the various phenotypes observed, there are a higher proteolysis capacity in ST59/ST398, higher expression of RNAlII and psm, better recognition from neutrophils, more apoptosis and better NETs degradation. Altogether, these results demonstrated that ST59 and ST398 provide accentuated virulence that make them proficient to fight against immune responses such as neutrophils NETs.

This study presents a good analysis of the epidemiology of S. aureus in China in hospital in the past few years.

Here are some general comments:

- 1) Lines 132-133: you're looking at proteases, leukocidins and even psm, however the enterotoxins (superantigens) are limited to ecg in Fig 2. Can you comment on prevalence of superantigen not in the ecg in the different lineages?
- 2) Fig 1: the choice of colors should be thought over, the difference in between 2 tones of blue, red or purple is not distinguishable for the reader
- 3) Line 217: Replace "Fruther" by "Further"
- 4) Lines 222-229: I would move this section earlier in the results, most of your results are more exciting and it feels like you end the results a bit flat, I would suggest to add it in the results section for Fig4 or even at Fig3 as you grew your strains in TSB for the proteolysis assay. Also, the growth defect normally observed in ST239 and ST5 is important to discuss in the case of the bacteria survival to neutrophils in Fig5A: we see a better survival for these 2 lineages
- 5) Line 238: Replace "." by "."
- 6) Line 284: Replace "miacrobial" by "microbial"
- 7) Line 285: Replace "neutrophils" by "neutrophils"
- 8) In the discussion, you should identify your figures: some of them are identified, but there were more figures discussed than what is identified
- 9) Lines 417-422: All the statistical methods should also be part of the other method section to identify which statistical method was used for which experimental method. It will be also be good as well to add it to the figure legends for an easier reading

Dear Editor and Reviewers:

Thank you for taking time out of your busy schedule to review the manuscript. We
have revised the manuscript carefully in accordance with your suggestions. The
revisions have been made as follows:

Author response to Reviewers #1

The manuscript by Chen et. al., "Phenotypic and Genomic Comparison of
Staphylococcus aureus Highlights Virulence and Host Adaptation Favoring the
Success of Epidemic Clones" described the genomic comparison of epidemic clones
by genomic analysis. In addition, the author also confirmed the high virulence of
ST59, ST398 and ST5-MSSA using vitro assays. Overall, I am impressed by the
organization of number of new information regarding S. aureus. I also have couples of
concerns, esp., no dominant clone identified in the MSSA strains. For clinical MSSA
strains, several clones including ST5, ST398, ST188, ST88, ST7 and others comprise
most of them, however, the ST59 clone which is identified as the dominant MRSA
clone in China, accounted for not such a big proportion in MSSA. The MSSA lineages
show considerable genetic diversity compared to the MRSA. For MRSA strains, there
are indeed dominant clones. Moreover, the proportion of dominant clones in MRSA is
much higher than that of other clones, and there are different dominant clones in
different regions. Therefore, the authors should focus the main topic of this study on
MRSA strains, due to there is no definition about the dominant clone in MSSA to
date.

**Response:** We thank the reviewer for their comments and constructive suggestions
regarding our paper.

**Abstract:**

1. Line 34-36: please delete the content of MSSA, and the enhanced proteolytic ability
needs to be verified by experiments.

**Response:**

We thank the reviewer for this comment. Just as the reviewer said, MRSA is
defined as clones rather than MSSA, the dominant clone in MSSA has not been
defined yet¹. The reason for the inclusion of MSSA is its higher virulence and the
decreasing rate of MRSA in China. We agree with the reviewer in deleting the content
of MSSA.

For the proteolysis assay, given the high number of isolates, we decided to use a
TSA plate containing 2% skim milk to measure proteolytic ability. This is an efficient
assay that has been applied in published papers^{2,3}. Moreover, the difference was
significant as the figure below shows.

Author response image 1

2.Line 39-40: please re-write "Besides differences in virulence".

**Response:** We agree with the reviewer. This phrase has been deleted from the revised

manuscript, and we have re-write this part in line 33.

3.Line 51-52: there are no dominant clones in MSSA strains.

**Response:** We agree with the reviewer. As our study focuses on the four STs, we have
decided to mainly talk about the MRSA clone. To clarify, the statement of dominant
clones in MSSA strains has all been deleted in the new manuscript.

4.Do an English revision

**Response:** Thank you for the suggestion, the English revision has been made.

**Introduction**

1.Line 65-67: This sentence is mainly aimed at MRSA and does not apply in MSSA.
The object of this study should be narrowed down to MRSA, not the whole S. aureus
group.

**Response:** We agree with the reviewer. This sentence has been adjusted in the revised
manuscript.

2.Line 70-71: ST398 has only been considered as CA-MRSA in the recent years,
which is mainly based on related clinical data. In fact, most ST398 strains are
considered as LA-MRSA.

Misconception.

**Response:** Thank you for pointing this out, we apologize for the misconception of
ST398.

3.Line 81-82: The chp gene mainly participates in promoting immune evasion, i.e.,
helping bacteria escape from immune recognition, thus persisting in the host. Why
does the chp gene enhance the cytolytic ability of ST59 clones?

**Response:** Thank you for raising this question. In earlier research, our association
study identified the chemotaxis inhibitory protein (chp) gene as a strong candidate for
involvement in the increased virulence potential of ST59. The Δ chp mutants showed a
higher level of impairment in cytolytic capacity than the wildtype strains⁴. As such,
the relationship between chp gene and cytolytic ability is under investigation.

4.please re-write line 82-85 to enhance its connection with the context.

**Response:** This sentence has been re-written in line 76-78 in the revised manuscript.

**Results**

1.Line 102-116: Which data is correct: 2013-2022 or 2020? Correct in all the text.

**Response:** We apologize for the wrong description. The correct date is 2013–2020.
We have corrected it in line 94.

Line 114: the term "community clone" should be defined by clinical data rather ST
types.

**Response:** Thank you for raising this point. We have deleted the term "community
clone" in the revised manuscript to reduce confusion.

2. Line 119: without phylogeographic and epidemiological data to support these
statements, I don't think the present Figure give us any information to support this
conclusion.

**Response:** Thank you for this comment, we agree with reviewer that we lacked the
phylogeographic and epidemiological data to prove frequent exchanges among
regions and hospitals. We have thus changed the sentence to “the phylogenetic
structure could not be fully explained by geographic sampling in either ST398 or
ST59 clones, with isolates collected from hospitals located in different Chinese cities
interspersed in the phylogeny” in lines 111-114.

3.Line 128: Reasons for inclusion of ST5?

**Response:** We thank the reviewer for this question. In our epidemiological
investigation across China, we found that the prevalence of ST5 has been maintained
at a stable state. ST5-MRSA has a very high prevalence in Shanghai and Zhejiang
provinces as others have previously reported⁵. According to Li et al, the ST5 MSSA
subtype has displayed a high virulence and enhanced magnitude in the past decades
and poses a serious threat in clinical *S. aureus* infections⁶. Hence, more attention
should be paid in the prevention and control of methicillin susceptible isolates.
Therefore, we think it is important to include an ST5-MRSA clone and ST5-MSSA
subtype into our analysis.

4. Line133: pvl was seldom found in ST59-MRSA and ST398-MSSA. I don't
understand the meaning of this sentence. Do you want to express that the pvl gene
exists in ST59 and ST398- MSSA, but it is not common in ST59 and ST398?

**Response:** We apologize for the lack of clarity in the text. The overall prevalence of
the pvl gene in our strain collections (n=142) is 5.63%, including five ST59-MRSA,
two ST398-MSSA, and one ST398-MRSA. We have changed the sentence in the
revised manuscript (line 128-129) for clarity.

5. Line 136-137: Interestingly, ST59, ST398-MRSA, and ST5-MSSA exhibited a
similar pattern to IEC type B. What is similar pattern?

**Response:** We thank the reviewer for pointing to this. The similar pattern refers to
immune evasion cluster (IEC) type B. IEC variants were discovered carrying different
combinations of sea (or sep), sak, chp, and scn. The B- type referred to carrying sak,
chp, and scn. The different IEC patterns among the four clones have been listed in
table S4.

6. Line 141-145: The paragraph is not clear, it is not understood. This paragraph is
confusing. Since we are exploring the difference between dominant clones and
non-dominant clones. Why do you compare ST398 with ST59, which is also a
dominant clone. Why do you compare ST5- MRSA with ST5-MSSA? Please reselect
clones to compare.

**Response:** Thank you for raising this question. This paragraph has been re-written to
focus on the comparison between the dominant clones and non-dominant clones at
line 140-143.

7. Line 174-176: The results indicate that ST239- and ST5-MRSA showed stronger
resistance to PMN (Polymorphonuclear leukocytes) mediated killing. If ST239 and
ST5 had stronger resistance to neutrophils, ST239 and ST5 should have stronger

capacity to survive in the host. The immune evasion ability is stronger, and it is not
easy to be killed by immune cells. It means the stronger ability of ST239 and ST5 to
persist in the host. However, the author stated that ST239 and ST5 are not the
dominant clones of BSIs. How do the authors explain this result?

**Response:** Thank you for raising this question. The neutrophil killing assay was
performed by incubating bacteria and neutrophils for 30 minutes. Since ST59 and
ST398 were more virulent, it is likely that they are more easily recognized by host
immune cells such as neutrophils, thus provoking a stronger immune response such as
bacterial killing in a short time. This might account for high bacterial mortality in
ST59 and ST398. However, this stronger immune response might cause inflammatory
reactions such as NETs formation, apoptosis, and occurrence of infectious diseases,
which makes them the epidemic clone. However, we observed the growth defect in
ST239 and ST5; thus, the long-term survival rate in the host may not be achieved.

8. Line 185-196: Again, why compared ST5-MSSA with ST5-MRSA. The results
shows that ST5- MSSA is the dominant clone of MSSA and it is more virulent. And
ST5-MRSA is not a dominant clone, so if ST5-MRSA is weak in virulence. However,
ST5-MRSA has a very high prevalence in Shanghai and Zhejiang provinces. If the
high virulence account for the prevalence of the ST5- MSSA, what account for the
ST5-MRSA in east China? How does the author explain it?

**Response:** We agree with the reviewer that ST5-MSSA is not an *S. aureus* clone.
However, for the reasons indicated above, we want to pay some attention to the
ST5-MSSA subtype. Since we want to compare the major STs that cause epidemics in
China, ST5 is non-negligible. Moreover, the reason for the popularity of ST5-MRSA
in east China is probably its antimicrobial resistance and adaptation to hospital
environment, which needs further study.

9. Line 217-220: Could you describe a little more for details on this part of results?

The title is ST398 shows higher adhesion and invasion capacity with keratinocytes.

But there is very little in this section.

**Response:** We thank the reviewer for this suggestion. We have changed the title of
this paragraph to match its content. The title has been edited to “Adhesion-associated
genotypic and phenotypic study of the different STs” in the revised manuscript.

**Discussion**

Line 235: *paiB*, *cfim*, *hysAVSaβ*, and *hly* replace "might contribute to ..." with
"might be associated with ...". Do these genes really account for the success of
epidemic clone? Actually, we don't know.

**Response:** We agree with the reviewer. The contributions of *paiB*, *cfim*, *hysA^{VSaβ}*, and
*hly* to the formation of epidemic clones need to be functionally verified. This will be
the focus of our future research. We have mentioned this again in the revised
manuscript in line 235-237.

Line 238: deleted ' ."

**Response:** We apologize for the format error.

Line 242-244: Again, MSSA does not have dominant or epidemic clones. The
definition of dominant and epidemic clones is limited to MRSA.

**Response:** We thank the reviewer for this comment. We have changed this sentence to
“ To our knowledge, this is the first study to compare differences at the genomic and
host cell-pathogen interaction phenotypic levels for the four MRSA clones” in the
revised manuscript in line 242-243 .

**M&M**

Line 331-332: How the author selected the strains?

**Response:** The strain selection process was illustrated in Fig S1. In short, we selected
142 strains comprising the major STs in our recent strain collections from 2019–2020.
Proteolysis and cytolytic assays were performed for the 142 strains. To compare host
cell-pathogen interactions for the different STs, 32 strains were selected based on 1)
spa type, SCC*mec* type, and virulence genes distribution; 2) representative phenotypic
characteristics; 3) relatively wide geographical distribution.

**Author response to Reviewers #2**

Chen et al studied the genetic and virulence phenotypes in between previously *S.*
*aureus*

epidemic lineages ST5 and ST239 compared to the now successful ST59 and ST398.

Among the various phenotypes observed, there are a higher proteolysis capacity in
ST59/ST398, higher expression of RNAlII and *psm*, better recognition from
neutrophils, more apoptosis and better NETs degradation. Altogether, these results
demonstrated that ST59 and ST398 provide accentuated virulence that make them
proficient to fight against immune responses such as neutrophils NETs.

This study presents a good analysis of the epidemiology of *S. aureus* in China in
hospital in the past few years.

**Response:** We appreciate the reviewer's insightful comments and recognition of our
work.

1) Lines 132-133: you're looking at proteases, leukocidins and even psm, however the
enterotoxins (superantigens) are limited to ecg in Fig 2. Can you comment on
prevalence of superantigen not in the ecg in the different lineages?

**Response:** Thank you for the suggestion. We have investigated other superantigens
besides the enterotoxin gene cluster (ecg) located within the VSa β genomic island,
namely tsst-1, sea, seb, sec, selk, selq, and so on. We have added the result of this part
to Fig 2 in the revised manuscript. And the description was added in line 129-131 in
the revised manuscript.

2) Fig 1: the choice of colors should be thought over, the difference in between 2
tones of blue, red or purple is not distinguishable for the reader.

**Response:** Thank you for the suggestion about the choice of color, we have changed
the color of Fig 1 to be one that is more distinguishable.

3) Line 217: Replace “Fruther” by “Further”

**Response:** We apologize for the incorrect spelling; the correct spelling has been
included.

4) Lines 222-229: I would move this section earlier in the results, most of your results
are more exciting and it feels like you end the results a bit flat, I would suggest to add
it in the results section for Fig4 or even at Fig3 as you grew your strains in TSB for
the proteolysis assay. Also, the growth defect normally observed in ST239 and ST5 is

important to discuss in the case of the bacteria survival to neutrophils in Fig5A: we
see a better survival for these 2 lineages.

**Response:**

Thank you for the useful suggestions. We have made the required changes to Fig 4.

In Fig 5A, the neutrophil killing assay was performed by incubating bacteria and
neutrophils for 30 min. Since ST59 and ST398 were more virulent, it is likely that
they are more easily recognized by host immune cells such as neutrophils, thus
provoking a stronger immune response such as bacterial killing in a short time. This
might account for the higher bacterial mortality in ST59 and ST398. We also observed
a growth defect in ST239 and ST5 (Fig 4); thus, the long-term survival rate in the host
may not be achieved. We have added this to the revised manuscript in the section on
bacteria survival against neutrophils.

5) Line 238: Replace ".." by "."

**Response:** We apologize for the format error.

6) Line 284: Replace "miacrobial" by "microbial"

**Response:** We apologize for the spelling error. The correct spelling has been inserted
instead.

7) Line 285: Replace "neutrophils" by "neutrophils"

**Response:** We apologize for the spelling error. The correct spelling has been inserted
instead.

8) In the discussion, you should identify your figures: some of them are identified, but
there were more figures discussed than what is identified.

**Response:** We thank the reviewer for the suggestions. We have identified all the
figures in the discussion section of the revised manuscript.

9) Lines 417-422: All the statistical methods should also be part of the other method
section to identify which statistical method was used for which experimental method.
It will be also be good as well to add it to the figure legends for an easier reading.

**Response:** We thank the reviewer for the suggestions. The statistical tools used for
each method have been added to the corresponding method section and figure
legends.

Thank you again to the reviewers and editors for your hard work! Please do not
hesitate to contact us if there are any questions.

Best wishes to you!

**Reference**

1. Planet, P. J. *et al.* Architecture of a Species: Phylogenomics of *Staphylococcus*
*aureus*. *Trends Microbiol.* **25**, 153–166 (2017).

2. Petrie, L. E., Leonard, A. C., Murphy, J. & Cox, G. Development and
validation of a high-throughput whole cell assay to investigate *Staphylococcus*
*aureus* adhesion to host ligands. *J. Biol. Chem.* **295**, 16700–16712 (2020).

3. Espadinha, D. *et al.* Distinct phenotypic and genomic signatures underlie
contrasting pathogenic potential of *staphylococcus epidermidis* clonal lineages.
*Front. Microbiol.* **10**, 1–17 (2019).

- 4. Chen, H. *et al.* Drivers of methicillin-resistant *Staphylococcus aureus* (MRSA)
lineage replacement in China. *Genome Med.* **13**, 1–14 (2021).
- 5. Jin, Y. *et al.* Genomic Epidemiology and Characterization of
Methicillin-Resistant *Staphylococcus aureus* from Bloodstream Infections in
China . *mSystems* **6**, (2021).
- 6. Jian, Y. *et al.* Increasing prevalence of hypervirulent ST5 methicillin
susceptible *Staphylococcus aureus* subtype poses a serious clinical threat.
*Emerg. Microbes Infect.* **10**, 109–122 (2021).

October 26, 2022

Prof. Hui Wang
Department of Clinical Laboratory, Peking University People's Hospital, Beijing, China
Department of Clinical Laboratory
No.11 Xizhimen South Street
beijing 100044
China

Re: mSystems00831-22R1 (Phenotypic and Genomic Comparison of Staphylococcus aureus Highlights Virulence and Host Adaptation Favoring the Success of Epidemic Clones)

Dear Prof. Hui Wang:

Thank you for submitting your article to mSystems. Your manuscript has been accepted, and I am forwarding it to the ASM Journals Department for publication. For your reference, ASM Journals' address is given below. Before it can be scheduled for publication, your manuscript will be checked by the mSystems production staff to make sure that all elements meet the technical requirements for publication. They will contact you if anything needs to be revised before copyediting and production can begin. Otherwise, you will be notified when your proofs are ready to be viewed.

Publication Fees:

If you would like to submit a potential Featured Image, please email a file and a short legend to msystems@asmusa.org. Please note that we can only consider images that (i) the authors created or own and (ii) have not been previously published. By submitting, you agree that the image can be used under the same terms as the published article. File requirements: square dimensions (4" x 4"), 300 dpi resolution, RGB colorspace, TIF file format.

We recognize that the video files can become quite large, and so to avoid quality loss ASM suggests sending the video file via <https://www.wetransfer.com/>. When you have a final version of the video and the still ready to share, please send it to mSystems staff at msystems@asmusa.org.

Sincerely,

Zackery Bulman
Editor, mSystems

Journals Department
table S3: Accept
table S1: Accept
figS1: Accept
figS3: Accept
figS2: Accept
table S2: Accept
data set 1: Accept